# Time to Assess Bias in Machine Learning Models for Credit Decisions

**Liming Brotcke**

Model Validation Group, Ally Financial, Charlotte, NC 28202, USA; liming.brotcke2@ally.com

**Abstract:** Focus on fair lending has become more intensified recently as bank and non-bank lenders apply artificial-intelligence (AI)-based credit determination approaches. The data analytics technique behind AI and machine learning (ML) has proven to be powerful in many application areas. However, ML can be less transparent and explainable than traditional regression models, which may raise unique questions about its compliance with fair lending laws. ML may also reduce potential for discrimination, by reducing discretionary and judgmental decisions. As financial institutions continue to explore ML applications in loan underwriting and pricing, the fair lending assessments typically led by compliance and legal functions will likely continue to evolve. In this paper, the author discusses unique considerations around ML in the existing fair lending risk assessment practice for underwriting and pricing models and proposes consideration of additional evaluations to be added in the present practice.

**Keywords:** ML; algorithm; fair lending; disparate; bias; discrimination

## 1. Introduction

Major advances in ML in the last few years have led to several tremendous break-throughs in the field of AI and extensive application of AI tools in a variety of fields and industries, including medical, manufacturing, education, judicial, and marketing. The financial sector is a relatively late adopter of ML and AI tools. Although ML algorithms have been applied to assist financial institutions in the areas of anti-money laundering, fraud detection, fraud monitoring, high-frequency trading, cyber risk monitoring, marketing, customer service, and credit scoring, the chosen ML techniques are far less complex. The most commonly used algorithms for core banking operations and compliance fulfillment include gradient boosting machines (GBM) and their variations, support vector machines (SVM), and random forests with a higher degree of explainability. In contrast, less transparent neural networks, deep learning methods, and NLP approaches are also seen in applications such as customer support and data and documentation processing. The modern era of the fintech industry post-Great Recession also drives non-bank lenders to leverage ML to increase their market share and gain sustained competitiveness. Many banks either partner with or acquire fintech firms to grow their portfolios and further increase ML use. When less transparent and explainable ML approaches are applied to develop credit scoring and pricing models, benefits are accompanied with new challenges, including around their comport with fair lending laws.

A wealth of literature has appeared on the subject of applicability of ML techniques to various banking operations, empirical analyses to test ML model strengths against classic regressions, AI interpretabilities, and fairness. This paper contributes to this effort by performing a review of the evolution of fair lending assessment, comparing nuances between regulatory focuses and bank practices, and discussing challenges posed by ML approaches to conventional fair assessment with proposed solutions. The focus of introducing alternative quantitative assessments is balanced with a healthy dose of fair lending and legal perspectives. This paper is organized as follows: Section 1 discusses why ML approaches

are preferrable to traditional statistical models. Sections 2 and 3 provide a brief discussion of fair lending laws and standard risk assessment performed by financial institutions and regulators. The author then highlights unique challenges presented by ML credit models to traditional risk assessment in Section 4, before offering suggestions on how to modify current practices acknowledging potential discriminatory risks associated with ML models in Section 5.

## 2. Machine Learning vs. Statistical Models

ML and traditional statistical models both leverage the same fundamental notions of probability. The former demonstrates notable advantages over the latter in several areas and is, therefore, favored by model developers. First, ML approaches are more adaptable and flexible when dealing with big data that include structured and unstructured data. Big data's size or type is beyond the ability of traditional relational databases to capture, manage, and process with low latency (IBM 2021). Big data sets are also rapidly generated and transmitted from a wide variety of sources. Processing and analyzing big data in a timely and economical manner requires AI and cognitive technologies that include computer vision, ML, natural language processing (NLP), speech recognition, and robotics. ML algorithms' ability to process unstructured data which conventional statistical models cannot helps ML algorithms generate more accurate predictions.

Second, ML models, by design, generate the most accurate predictions possible. Traditional regression analysis, with its most classic form of statistical models, is instead applied to make inferences about relationships between variables and is rooted in economic and other social science theories. To build a statistical regression, the following steps are generally taken by trained econometricians and statisticians:

- Identify the problem in theoretical and operational terms;
- Define the probability spaces, then collect data, design the study, and determine if sampling is needed;
- Generate hypotheses and explore data before developing the model following an iterative process that tests regression assumptions;
- Conduct model diagnostics to evaluate the individual and aggregate effect of explanatory variables on the target variable for each model version.

Building a statistical model involves constructing a mathematical description of observed real-world phenomena that account for the uncertainty and/or randomness involved in that system. The process is mathematics and labor-intensive. Statistical modeling is more about identifying relationships between the target and explanatory variables. Statistical models therefore can be expanded for prediction use only once the relationships are statistically proven to be sound via model statistics and other statistical testing.

In comparison, even though the theories behind ML are the same as mathematics and statistics, the development and implementation of ML relies on knowledge and concepts from other fields too, including optimization, matrix algebra, calculus, computer science, and engineering. As stated by BLDS LLC, Discover Financial Services, and H2O.ai (2020), AI and ML algorithms search for obscured patterns within and across features, enabling computers to learn to make decisions both faster and often with greater accuracy than can be achieved by humans. Since ML models are not necessarily rooted in economic or other social science theories, their value is not defined by their relationship-finding ability but rather on how well they predict. Put differently, relationship confirmation is not required when evaluating the strengths of ML techniques. Hence, measures other than accuracy are not prioritized in many industries when, for example, the objective of ML is to ensure the machine arm grabs a package from the same location every time and all the time. The interpretability of ML is less relevant when applying it to fields where accuracy and errors can be precisely defined and measured. Additionally, certain key developmental steps that are associated with statistical models, such as segmentation evaluation and variable treatment, become less necessary when building ML models.

Third, ML algorithms have a loose reliance on assumptions relative to statistical models. Statisticians often emphasize that statistical inference relies on model assumptions which need to be checked, although the rigor of assumption testing and reliance on testing results are an ongoing debate. Tijmstra (2018) believes that it is critical to test assumptions in order to establish the validity of inferences because only if the statistical model is specified (approximately) correctly can inferences about hypotheses of interest be relied upon. Mayo (2018) states that invalidity of model assumptions and the failure to check them is at least partly to blame for what is currently referenced as "replication crisis". On the contrary, Shamsudheen and Hennig (2021) conclude that either running a less constrained test or running the model-based test without preliminary testing have been found to be superior to the combined procedure involving a preliminary mis-specification test.

Performing model assumption testing has its advantage as it enables statistical models, a method of mathematically approximating the reality, to be developed on relatively small datasets that allow for human comprehension and interpretation of the established relationships, so long as model assumptions are generally valid. When enormous numbers of data become available, assumption testing becomes less critical because the law of large numbers (LLN) applies where the sample mean and the sample probability converge to the population mean and probability as the sample size grows. When working with massive data, along with matching computational and storage power of computers, ML tends to easily outperform regression analysis by identifying patterns in the data that go beyond standard correlation analysis. The ability to extract unseen and nonlinear features from massive data of ML algorithms make some researchers argue that the complex nature of ML provides opportunities to make models fairer (BLDS LLC, Discover Financial Services, and H2O.ai).

## 3. Fair Lending Laws and Types of Discrimination

The United States has two different federal laws that deal with discrimination in lending: The Fair Housing Act (FHA) and the Equal Credit Opportunity Act (ECOA). These fair lending laws prohibit lenders from discriminating in credit transactions on the basis of race, color, national origin, religion, sex, marital status, and age. Both laws protect consumers by prohibiting unfair and discriminatory practices.

The FHA was passed as part of the Civil Rights Act of 1968, which prohibits discrimination including but beyond just lending in many activities of the residential real estate industry based upon race, color, religion, sex, handicap, familial status (if a household includes children), and national origin. The 1974 ECOA lists a series of prohibited bases, including race, sex, national origin, and age, as well as less common factors such as whether the individual receives public assistance, that cannot be used in deciding whether to provide credit, the interest rate, or any other aspects of the credit transaction. Walter (1995) pointed out two other laws often mentioned in discussions of fair lending are the Home Mortgage Disclosure Act of 1975 (HMDA) and the Community Reinvestment Act of 1977 (CRA). Although both laws play a part in current fair lending enforcement, neither prohibits discriminatory lending; hence, neither is defined as a fair lending law according to Walter.

After the Great Recession, the Dodd–Frank Wall Street Reform and Consumer Protection Act of 2010 (Dodd–Frank Act) expanded the definition of unfair and deceptive acts. The Dodd–Frank Act makes it unlawful for any company which provides any financial products or services to consumers to engage in any acts or practices considered to be unfair, deceptive, or abusive ("UDAAP") (FDIC 2021). The Consumer Financial Protection Bureau (CFPB) enforces UDAAP. The Federal Trade Commission (FTC) and other prudential regulators (the Federal Reserve Board (FRB), the Federal Deposit Insurance Corporation (FDIC), and the Office of the Comptroller of the Currency (OCC)) enforce Unfair or Deceptive Acts and Practices (UDAP). In practice, an unfair, deceptive, or abusive act or practice may also violate other federal or state laws, such as the Truth in Lending Act (TILA), the Fair Credit Reporting Act, and the Fair Credit Billing Act.

Regulators take enforcement and supervisory actions for three types of lending discrimination under the ECOA and the FHA: overt discrimination, disparate treatment, and disparate impact. In practice, overt discrimination receives little regulatory attention with respect to fair lending assessment because modern regulated financial institutions seldom blatantly offer more favorable terms to one group versus another based solely on a prohibited factor, such as gender.

Disparate treatment can occur when lenders apply different or inconsistent treatment to loan applicants based on prohibited factors that are not fully explained by relevant, non-discriminatory factors (CFPB 2021). Disparate treatment includes redlining, which is a form of illegal disparate treatment in which a lender provides unequal access to credit, or unequal terms of credit, because of the race, color, national origin, or other prohibited characteristic(s) of the residents of the area in which the credit seeker resides or will reside or in which the residential property to be mortgaged is located. Redlining hence may violate both the FHA and the ECOA.

Of the three types of discrimination, disparate impact is the most scrutinized and most likely to be focused on by regulators in a fair lending review. Disparate treatment applies to a wide range of issues, such as pricing, underwriting, or steering. Financial institutions, therefore, are more likely to fall into this discrimination category because factors used to assist and drive lending and pricing decisions, including default risk likelihood and creditworthiness, by nature, will cause inconsistency.

Disparate impact occurs when any individuals receive equivalent treatment, but a lending policy has a disparate effect on a prohibited basis. As defined in the Interagency Fair Lending Examination Procedures, "when a lender applies a racially or otherwise neutral policy or practice equally to all credit applicants, but the policy or practice disproportionately excludes or burdens certain persons on a prohibited basis, the policy or practice is described as having a "disparate impact. (FFIEC 2021)" In comparison, disparate treatment is done intentionally, reflecting differences in policies, and hence produces inconsistent outcomes. Disparate impact tends to be unplanned, with consistent policies and applications. It is often easier to prove disparate treatment than disparate impact. Historically, the perceived controversy embedded in the definition of disparate impact also means regulators seldom use it as the sole discrimination target in most fair lending reviews and inquiries.

## 4. Traditional Fair Lending Assessment by Regulators and Bankers

Regulators have been conducting fair lending risk assessments to check for evidence of disparate treatment, redlining, and disparate impact as part of the compliance examination for over four decades. Since the late 1970s, banking regulatory agencies have also incorporated statistical tests to aid them in their search for evidence of discrimination related to mortgage lending (Shamsudheen and Hennig 2021), inspired by pioneer research work carried out by Cleary (1968), Thorndike (1971), and Darlington (1971). Specific statistical tests employed by examiners may vary but all are rudimentary with a focus on identifying if there is a correlation between minority status and the frequency of loan denial, holding other factors constant. When a potential fair lending violation correlation is identified, examiners will engage in manual review of a sample file of loan applications to further search for evidence of discrimination. The statistical testing procedure proposed by academia became more sophisticated and employed more creditworthiness variables in the 1990s, as summarized by Calem and Canner (1995), Bauer and Cromwell (1994), and Stengel and Glennon (1995). Since only one in ten banks receives enough mortgage loan applications to ensure the statistical validity of these tests, as detailed by Calem and Canner (1995), the enhanced statistical testing has a very limited use outside of the large bank community. Alternatively, informal qualitative techniques are more consistently applied by examiners to banks regardless of their sizes. Aside from analyzing sampled applications, regulators also evaluate an individual bank's lending decision criteria, compare the racial and ethnic makeup of loan applications, review booked loans against that of market coverage, and

review policy, procedures, and standards for the lending practices. Research completed by the Board of Governors of the Federal Reserve System in 2010 yielded no evidence of disparate impact by race or gender but revealed limited disparate impact by age, in which the use of variables related to an individual's credit history appear to lower the credit scores of older individuals and increase them for the young (Avery et al. 2012).

Compared to the limited statistical tests regulators use, large financial institutions tend to rely more on quantitative methods to detect presence of discrimination on protected classes due to underwriting and/or pricing decisions. Two basic types of statistics, descriptive statistics and inferential statistics, form the framework of fair lending according to Lindsey-Taliefero (2001). Descriptive statistics are used to summarize data and consist of frequency distributions; measure of central tendency including mean, median, and mode; measures of variability including range, variance, and standard deviation; and measure of association including proportions, odds, and odds ratios. Cross-tabulation along with chi-square and *t*-tests are commonly used to explore relationships between variables and test mean differences between groups.

Mean measuring of two independent groups is also referred to as effect size calculation (Becker 2000). Different scholars proposed different ways to measure the size of an effect. The most commonly used statistics include Cohen's d, Hedges' g, and Glass's delta (Cohen 1988; Glass et al. 1981). Cohen's d, also known as the standardized mean difference (SMD), is determined by calculating the mean difference between two groups, and then dividing the result by the pooled standard deviation. When two groups have similar standard deviations and are of the same size, Cohen's d is the appropriate effect size measure. Hedges' g is an alternative when there are different sample sizes. Hedges' g provides a bias correction (using the exact method) to Cohen's d for small sample sizes. For sample sizes >20, the results for both statistics are roughly equivalent. When the standard deviations are significantly different between two groups, Glass's delta is a better choice. Glass's delta is defined as the mean difference between the experimental and control group divided by the standard deviation of the control group.

Inferential statistics are regression analysis in which conclusions and inferences are drawn from the data and thereby uncertainty is quantified. By assessing different factors and/or variables in aggregates, such as the debt-to-income ratio (DTI), loan to value (LTV), and FICO, regression analysis helps identify causes and account for discrepancies in order to explain decisioning and pricing disparities. Linear and logistic regressions are the two main techniques. A linear regression uses the ordinary least square (OLS) method to estimate the regression equation which describes how a dependent variable is related to one or more independent variables. For example, one can build a linear regression to quantify how the annual percentage rate (APR) or discretionary component of pricing can be explained by LTV and FICO and use $R^2$, *F*-statistics, and *T*-statistics to make inferences. When the objective is to assess if there is any discrimination in the dichotomous decision, such as loan approval or deny decision, a logistic regression analysis can be resorted. A logistic regression is approximated via the maximum likelihood estimation (MLE). Pseudo $R^2$ classification measures such as C-index are often employed to draw conclusions. Royston and Altman (2010) advocate using simple graphics, such as a histogram or dot plot of the risk score in the outcome groups, to provide further insight into discrimination, along with discussion of the comparative merits of the c-index and the effect size, namely, difference in risk score, between the outcome groups. They also illustrate an alternative overlap measure that uses the area under the minimum of the two density functions to suggest the degree of discrimination, i.e., larger overlap implies weaker discrimination (Royston and Altman 2010).

Alternative approaches have also been explored and tested in the last decade. Several large and small banks implemented the BISG (Bayesian Improved Surname Geocoding) method developed by the RAND Corporation to help U.S. organizations produce accurate, cost-effective estimates of racial and ethnic disparities within datasets and illuminate areas for improvement. Researchers have applied Bayes' Rule to predict the race/ethnicity of

an individual using the individual's surname and geocode location, such as (Elliott et al. 2008; Elliott et al. 2009; Imai and Khanna 2016). However, the accuracy and reliance of BISG estimates when applied to point estimates instead of aggregate data remain controversial topics among banking practitioners and regulators, resulting in limited use in practice. Voicu (2018), Director of the Compliance Risk Analysis Division of the Office of the Comptroller of the Currency (OCC), validated the new Bayesian Improved First Name Surname Geocoding (BIFSG) method using first name information on a large sample of mortgage applicants who self-report their race/ethnicity and claimed that BIFSG outperforms BISG, in terms of accuracy and coverage, for all major racial/ethnic categories.

In recent years, a few banks have also leveraged ML to determine the feasibility of predicting complaint resolutions because complaints from individuals who believe they are subject to prohibited credit discrimination will lead to investigations by enforcement agencies. For instance, the text mining technique and K "Nearest Neighbor" Classification (KNN) are applied to facilitate proper complaint review priority. By mathematically mapping data to Euclidian distances, class assignments are made by proximity to nearest neighboring data points. Unlike a credit scoring model developed with an ML algorithm, this type of ML application does not contribute to potential discriminatory treatment. Instead, it is applied to optimize the processing of complaints.

## 5. ML Techniques and Challenges to Traditional Fair Lending Assessment

Recent innovations, such as incorporating alternative data to establish credit criteria and the use of ML techniques, have promoted higher reliance in automated credit decisions, and hence reduce the use of discretionary and judgmental overrides which could result in bias and discriminatory loan approval and pricing decisions. Meanwhile, the growing use of ML and alternative data also creates new fair lending risks and challenges the appropriateness and completeness of traditional fair lending assessment approaches.

When it comes down to loan origination decisions, the most sought-after ML techniques by financial institutions are XGBoost and Light GBM because of their relatively simple theoretical setup and high comparability with logistic regression. Empirical applications of other approaches, such as random forest, neural networks, and SVM, outside academia are very limited for practical concerns around increased difficulty for interpretability and adverse action reason code (AARC) generation. The ability of ML to handle a massive number of data makes it easier for bias embedded in data to impact algorithmic decisions unnoticed. Regulations and laws prohibit overt and intentional discrimination from using direct and close proxies of prohibited bases. It is well-known by bankers that age, gender, ethnicity, and national origin should be excluded from the candidate variables when developing underwriting policies and scoring and pricing models. There is, however, no consensus on what qualifies as a close proxy. In the financial regulatory sector, if a variable's predictive power solely or largely attributes its correlation with a prohibited basis, say, a zip code, then the variable is a close proxy. Traditional credit scoring models used to assist loan origination and pricing are largely parametric, which means they fall into a specific family of probability distributions and can, hence, be estimated via statistical regressions.

Model developers at leading, large financial institutions typically have established procedures and a mature process to guide variable selection during model development to avoid including close proxies of protected classes in a model. Data other than raw and derived credit bureau information, specific loan terms for closed-end products, and application information collected from the underwriting stage are seldom allowed to come into the model and the credit decision process. Developers' efforts to eliminate bias in models via rigorous variable selection are further verified and confirmed by second-line validators and fair lending specialists before the model is approved for use. Evaluation of statistical significance and signs of regression model coefficients is an effective way to assess the disparate impact concern. Although the existing approaches do not guarantee

elimination of discriminatory practice, they have proven to be reliable when dealing with regression models without identifying major deficiencies.

The specific concern of disparate impact raised by credit scoring models that the independent variables may unfairly disadvantage particular populations does not go away when such models are developed with ML algorithms. Most ML models cannot be depicted in the form of regression equations. When applying non-parametric ML algorithms for the same purpose, the parameters are in infinite-dimensional parameter spaces that are either distribution-free or have a specified distribution with unspecified parameters. Individual variables that could serve as proxies for prohibited bases can be harder to identify in ML algorithms, compared to in the traditional models. Additionally, bias embedded in historical data where loan approval and pricing decisions could reflect legacy racial, gender, or age discriminatory practices, could be replicated and manifested by neutral algorithms. For instance, if the model data include small business ownership and applicants' school and degree, an ML algorithm will select those features if inclusion of such information improves the prediction accuracy, even if the combination of such data could be perceived as a proxy of prohibited bases. Put differently, a disparate impact could be more likely to occur in non-parametric ML models. Although qualified quantitative analysts and data scientists with full access to the model can explain how the algorithms work, it is more complex to explain how individual decisions are made. The less transparent nature of ML models further elevates regulatory concern on banks' fair lending practice.

Without model specifications, compliance specialists cannot review independent variables and coefficients for ML models as with regression models. This implies that data used to drive the loan approval/decline decision could be close proxies that would not have survived in regression models. This concern is particularly relevant when alternative data are also included to increase prediction accuracy. Unlike traditional financial data from the credit bureaus, there are more unknowns and uncertainties about alternative data and their impact on decisions. Although coefficients and feature importance scores both assess the contribution of a variable in prediction, it is worth noting that the former is a direct measure of variable importance, while the latter is not and requires additional analysis. The feature importance measure automatically takes into account all interactions with other features, which makes is hard to quantify the contribution of individual features. To isolate the importance of individual features, quantitative and computer science skills are needed, which most compliance and legal teams do not possess.

The fairness concern could be further exacerbated if banks purchase credit ML models offered by vendor solutions and do not have adequate opportunity to review the ML models. To properly train an ML algorithm, developers follow a process similar to building regression models with heavy dependence on data. During the learning phase, training data constitutes the comprehensive representation of the real world, which the algorithm seeks to approximate. If the training data include any kind of unwanted bias, the resulting algorithm will incorporate and enforce it. To mitigate this concern, developers must have a profound understanding of input and output variables, as well as robust knowledge of the mechanism of the chosen algorithm. As of today, this is certainly not true for all vendors and fintech firms serving the financial industry. As a result, some third-party solutions could make neutral ML algorithms that produce biased predictions that only developers with the necessary market knowledge and technical proficiency can recognize. Furthermore, if the bank does not have direct insight into the feature selection criteria, the ML algorithm's predictability could also come at the cost of severe model overfitting, as well as decisions on prohibited bases.

Algorithm fairness also attracts the attention of regulators and researchers in AI, software engineering, and law communities. Verma and Rubin (2018) summarized prominent notions of fairness and numerous definitions of fair treatment proposed in the last few years. Gajane and Pechenizkiy (2018) surveyed how fairness is formalized in the machine learning literature for the task of prediction and presented these formalizations with their corresponding notions of distributive justice from the social sciences literature. Most defi-

nitions are related to either group fairness, including demographic parity, predictive rate parity, and equalized odds; individual fairness; unawareness; or counterfactual fairness. A confusion matrix, also known as an error matrix, allows visualization of the performance of an algorithm and is a popular measure used for solving classification problems. Information in rows and columns of the matrix not only describes the accuracy of a classification model, but also serves as a statistical measure of fairness. Advancement in academia makes it possible to identify and even potentially quantify fair lending risk stemming from alternative data and ML algorithms. However, it requires appropriate quantitative skills.

## 6. Modify Fair Lending Assessment Approach and Process

Bank credit underwriting systems, supported by conventional regression models or ML algorithms, are subject to fair lending laws and regulations. The prevalent approach for evaluating the compliance of credit underwriting models with fair lending laws consists of pre-implementation review and post-implementation monitoring. After a model is developed, Compliance reviews model inputs along with design choices and conducts pre-implementation fair lending testing. Post-model implementation, Compliance performs ongoing monitoring and periodic back testing of model outcomes and trend analyses. Although the two-phase risk assessment approach applies to all models, in theory, the traditional methods of assessing fair lending risk, in practice, face substantive challenges when ML models are developed for loan approval and pricing decisions. The current risk assessment is incapable of identifying bias in credit and pricing decisions based on ML models due to the absence of regression equations and independent variables, and the lack of input transparency from alternative data.

The conventional fair lending assessment continues to have merit and should apply to ML models. Meanwhile, new approaches should be explored, tested, and incorporated into a firm's oversight of ML models. For instance, traditional metrics, such as marginal effect, adverse impact ratio (AIR), and standardized mean difference (SMD), with a focus on evaluating model outcome differences across groups, can be enhanced by leveraging other bias measures specific to ML and the AI system it supports.

One approach is to leverage the confusion matrix, which can be used to measure model accuracy and fairness. Different metrics can be derived from the basic confusion matrix and used to assess different fairness definitions, including demographic parity, equal selection parity, predictive parity, equalized odds, etc. Fairness based upon predicted outcome is different from fairness based on predicted and actual outcomes. It is also different from fairness based on predicate probabilities and actual outcome. It is important to match the fairness definition with the appropriate calculation.

Another common method banks have applied is to leverage AI interpretability measures. Enhanced performance of AI is achieved through increased ML model complexity but comes at the cost of transparency. Explainable AI therefore becomes a field of interest for computer science and mathematics that are concerned with the development of new methods that explain and interpret ML models. Global model interpretability helps us to understand the distribution of your target outcome based on the features. Local interpretability methods can be applied to a single prediction or a group of predictions. Researchers have identified four major categories for interpretability methods. They are summarized by Linardatos et al. (2020) as methods for explaining complex black-box models, methods for creating white-box models, methods that promote fairness and restrict the existence of discrimination, and methods for analyzing the sensitivity of model predictions. The financial industry has adopted a few approaches, including the Integrated Gradients approach, Class Activation Maps (CAM), LIME, SHAP, and Counterfactuals. Those measures not only help interpret the algorithms but also shed light on a classifier's fairness.

It is important to note that none of those methods directly address data bias, which could be the result of errors in collecting and recording data, a sampling selection problem that misrepresents the population, or data with historical human biases embedded. A transparent and repeatable data collection process with built-in quality controls and

checks, along with enhanced training, can largely eliminate human and some system errors. Random sampling and simple regression tests, such as Heckman's test and the Lagrange multiplier test, are known methods to avoid and test sample selection bias. Historical human bias, however, is difficult to detect and correct, and prone to be replicated and even amplified by ML techniques. To mitigate this bias, model developers must possess profound knowledge of credit and lending in order to perform a deep review of key features of the ML models. Inexperienced modelers oftentimes lack this ability to identify and challenge this type of bias because it takes years of practice to build that level of expertise.

Aside from the technical proficiency, the maturity of a bank's ML governance also influences the level and depth of additional statistical testing on fairness. It also determines if new fair lending assessment processes should be established, or if existing processes should be modified. The process enhancement may also trigger updates to policies and procedures for the fair lending risk assessment framework.

The most effective way to evaluate lending-related unfairness is to compare model-based decisions across different gender and racial groups. Since creditors are generally prohibited from collecting most types of prohibited basis data, except in the context of mortgage lending, it has been quite challenging for lenders of other consumer products, including auto loans, credit cards, and personal loans, to directly compare approved/declined loans and pricing across groups. This data limitation becomes more acute when dealing with ML models. When proper data and risk control are in place, banks may be able to leverage vendor data with self-reported gender, race, and ethnicity to assist the fair lending assessment. LexisNexis is one of the vendors that regularly captures such information in their data, which is supplied to most deposit business. For banks that offer mortgages along with other forms of credit, those banks may be able to merge mortgage and other data. If the mortgage sample size has a large enough overlap with other customer data, regression analysis can be included in the fair lending assessment aside from direct outcome evaluation. Although not a perfect solution, the overall accuracy might still be greater than estimating an individual's race and ethnicity with administrative records, as seen in the BISG approach. Other information such as zip codes can be given a fresh look and considered when engaging in aggregate fair lending risk assessment that stems from creditworthiness differentiation. Rigorous data assessment of non-financial alternative data should be conducted as well to gain insights into their association with potential unfair treatment.

Overall, financial institutions with active ML models in credit lending have taken proactive actions to properly identify and mitigate potential increases in fair lending risk. Although solutions for addressing identified discrimination are yet to be finalized, evaluating the completeness and appropriateness of existing risk assessment practices, including key stakeholder roles and responsibilities, is necessary for banks who are new to this space. This evaluation could amount to a new risk assessment of fair lending oversight of ML models that ensures a firm's continued compliance with fair lending regulation.

**Funding:** This research received no external funding. The APC was funded by Ally Financial.

**Data Availability Statement:** Not applicable.

**Conflicts of Interest:** The author declares no conflict of interest.

**Disclaimer:** The views presented in this research are solely those of the author and do not necessarily represent those of Ally Financial Inc (AFI) or any subsidiaries of AFI.

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
