# Peer review of "Time to Assess Bias in Machine Learning Models for Credit Decisions"

_jrfm, doi:10.3390/jrfm15040165_

Round 1

Reviewer 1 Report

In this study, the authors described the advantages of ML over traditional stastistical methods, and ML still has some problems need to be solved. I have one main problem:

The author mentioned that "ML’s ability to handle massive data makes it easier for bias embedded in data to impact algorithmic decision unnoticed." In my opinion, this ability also makes ML outperform traditional methods based on statistics. Because ML can extract unseen and nonlinear features from massive data. Could the author provide any practical/potential suggestions or solutions that how to mitigate the bias embedded in data.

Author Response

Dear Sir./Madam,

Thank you very much for your review and feedback. Please see my response to your comment.

Point 1: Could the author provide any practical/potential suggestions or solutions that how to mitigate the bias embedded in data.

Response 1: This is a great ask and not an easy task. Bias embedded in data, which reflects the credit lending practice and history, is probably the root cause of unfair treatment in my opinion. ML algorithms simply reveal or manifest such biases. I updated Section V. Modify Fair Lending Assessment Approach and Process and stated clearly that the suggested methods do not directly mitigate data bias. Other approaches that I suggested focus on identifying potential unfairness in the model outcomes which is an indirect way of mitigating data bias.

Additionally, aside from addressing your comment, I also incorporated feedback from other viewers from my organization. An updated manuscript will be provided to you shortly.

I greatly appreciate your time and consideration.

Sincerely yours,

Liming Brotcke

Reviewer 2 Report

In this paper, the author discusses sources of ML challenges to the existing fair lending risk assessment practice for underwriting and pricing models and proposes additional evaluation to be added to address potential gap in the present practice. Comments to the authors: 1) Contributions of this paper are not clear. Mention major contributions of this paper at the end of the Introduction section. 2) Explain clearly the application of ML in loan underwriting and pricing? 3) What id the role of support vector machines (SVM) in the present paper? 4) What are the challenges involved in the ML applied in this paper?

Author Response

Dear Sr./Madam,

Thank you very much for your review and feedback. Please see the attachment for my response. Aside from addressing your comments (track changes version) I also incorporated feedback from other reviewers from my organization in the current version.

I greatly appreciate your time and consideration.

Sincerely yours,

Liming Brotcke

Round 2

Reviewer 2 Report

All the comments are addressed in the revised manuscript.